# Escaping from saddle points on Riemannian manifolds

**Yue Sun**
University of Washington
Seattle, WA 98105
yuesun@uw.edu

**Nicolas Flammarion**
EPFL
Lausanne, Switzerland
nicolas.flammarion@epfl.ch

**Maryam Fazel**
University of Washington
Seattle, WA 98105
mfazel@uw.edu

## Abstract

We consider minimizing a nonconvex, smooth function $f$ on a Riemannian manifold $\mathcal{M}$. We show that a perturbed version of Riemannian gradient descent algorithm converges to a second-order stationary point (and hence is able to *escape* saddle points on the manifold). The rate of convergence depends as $1/\epsilon^2$ on the accuracy $\epsilon$, which matches a rate known only for unconstrained smooth minimization. The convergence rate depends polylogarithmically on the manifold dimension $d$, hence is almost dimension-free. The rate also has a polynomial dependence on the parameters describing the curvature of the manifold and the smoothness of the function. While the unconstrained problem (Euclidean setting) is well-studied, our result is the first to prove such a rate for nonconvex, manifold-constrained problems.

## 1 Introduction

We consider minimizing a non-convex smooth function on a smooth manifold $\mathcal{M}$,

$$\text{minimize}_{x \in \mathcal{M}} \ f(x), \tag{1}$$

where $\mathcal{M}$ is a $d$-dimensional smooth manifold[1], and $f$ is twice differentiable, with a Hessian that is $\rho$-Lipschitz (assumptions are formalized in section 4). This framework includes a wide range of fundamental problems (often non-convex), such as PCA (Edelman et al., 1998), dictionary learning (Sun et al., 2017), low rank matrix completion (Boumal & Absil, 2011), and tensor factorization (Ishteva et al., 2011). Finding the global minimum to Eq. (1) is in general NP-hard; our goal is to find an approximate second order stationary point with first order optimization methods. We are interested in first-order methods because they are extremely prevalent in machine learning, partly because computing Hessians is often too costly. It is then important to understand how first-order methods fare when applied to nonconvex problems, and there has been a wave of recent interest on this topic since (Ge et al., 2015), as reviewed below.

In the Euclidean space, it is known that with random initialization, gradient descent avoids saddle points asymptotically (Pemantle, 1990; Lee et al., 2016). Lee et al. (2017) (section 5.5) show that this is also true on smooth manifolds, although the result is expressed in terms of nonstandard manifold smoothness measures. Also, importantly, this line of work does not give quantitative rates for the algorithm's behaviour near saddle points.

Du et al. (2017) show gradient descent can be *exponentially slow* in the presence of saddle points. To alleviate this phenomenon, it is shown that for a $\beta$-gradient Lipschitz, $\rho$-Hessian Lipschitz function, cubic regularization (Carmon & Duchi, 2017) and perturbed gradient descent (Ge et al., 2015; Jin et al., 2017a) converges to $(\epsilon, -\sqrt{\rho\epsilon})$ local minimum [2] in polynomial time, and momentum based method accelerates (Jin et al., 2017b). Much less is known about inequality constraints: Nouiehed et al. (2018) and Mokhtari et al. (2018) discuss second order convergence for general inequality-constrained problems, where they need an NP-hard subproblem (checking the co-positivity of a matrix) to admit a polynomial time approximation algorithm. However such an approximation exists only under very restrictive assumptions.

An orthogonal line of work is optimization on Riemannian manifolds. Absil et al. (2009) provide comprehensive background, showing how algorithms such as gradient descent, Newton and trust region methods can be implemented on Riemannian manifolds, together with asymptotic convergence guarantees to first order stationary points. Zhang & Sra (2016) provide global convergence guarantees for first order methods when optimizing geodesically convex functions. Bonnabel (2013) obtains the first asymptotic convergence result for stochastic gradient descent in this setting, which is further extended by Tripuraneni et al. (2018); Zhang et al. (2016); Khuzani & Li (2017). If the problem is non-convex, or the Riemannian Hessian is not positive definite, one can use second order methods to escape from saddle points. Boumal et al. (2016a) shows that Riemannian trust region method converges to a second order stationary point in polynomial time (see, also, Kasai & Mishra, 2018; Hu et al., 2018; Zhang & Zhang, 2018). But this method requires a Hessian oracle, whose complexity is $d$ times more than computing gradient. In Euclidean space, trust region subproblem can be sometimes solved via a Hessian-vector product oracle, whose complexity is about the same as computing gradients. Agarwal et al. (2018) discuss its implementation on Riemannian manifolds, but not clear about the complexity and sensitivity of Hessian vector product oracle on manifold.

The study of the convergence of gradient descent for non-convex Riemannian problems is previously done only in the Euclidean space by modeling the manifold with equality constraints. Ge et al. (2015, Appendix B) prove that stochastic projected gradient descent methods converge to second order stationary points in polynomial time (here the analysis is not geometric, and depends on the algebraic representation of the equality constraints). Sun & Fazel (2018) proves perturbed projected gradient descent converges with a comparable rate to the unconstrained setting (Jin et al., 2017a) (polylog in dimension). The paper applies projections from the ambient Euclidean space to the manifold and analyzes the iterations under the Euclidean metric. This approach loses the geometric perspective enabled by Riemannian optimization, and cannot explain convergence rates in terms of inherent quantities such as the sectional curvature of the manifold.

**Contributions.** We provide convergence guarantees for perturbed first order Riemannian optimization methods to seond-order stationary points (local minima). We prove that as long as the function is appropriately smooth and the manifold has bounded sectional curvature, a perturbed Riemannian gradient descent algorithm escapes (an approximate) saddle points with a rate of $1/\epsilon^2$, a polylog dependence on the dimension of the manifold (hence almost dimension-free), and a polynomial dependence on the smoothness and curvature parameters. This is the first result showing such a rate for Riemannian optimization, and the first to relate the rate to geometric parameters of the manifold.

Despite analogies with the unconstrained (Euclidean) analysis and with the Riemannian optimization literature, the technical challenge in our proof goes beyond combining two lines of work: we need to analyze the interaction between the first-order method and the second order structure of the manifold to obtain second-order convergence guarantees that depend on the manifold curvature. Unlike in Euclidean space, the curvature affects the Taylor approximation of gradient steps. On the other hand, unlike in the local rate analysis in first-order Riemannian optimization, our second-order analysis requires more refined properties of the manifold structure (whereas in prior work, first order oracle makes enough progress for a local convergence rate proof, see Lemma 1), and second order algorithms such as (Boumal et al., 2016a) use second order oracles (Hessian evaluation). See section 4 for further discussion.

## 2 Notation and Background

We consider a complete[3], smooth, $d$ dimensional Riemannian manifold $(\mathcal{M}, \mathfrak{g})$, equipped with a Riemannian metric $\mathfrak{g}$, and we denote by $\mathcal{T}_x\mathcal{M}$ its tangent space at $x \in \mathcal{M}$ (which is a vector space of dimension $d$). We also denote by $\mathbb{B}_x(r) = \{v \in \mathcal{T}_x\mathcal{M}, \|v\| \leq r\}$ the ball of radius $r$ in $\mathcal{T}_x\mathcal{M}$ centered at 0. At any point $x \in \mathcal{M}$, the metric $\mathfrak{g}$ induces a natural inner product on the tangent space denoted by $\langle \cdot, \cdot \rangle : \mathcal{T}_x\mathcal{M} \times \mathcal{T}_x\mathcal{M} \to \mathbb{R}$. We also consider the Levi-Civita connection $\nabla$ (Absil et al., 2009, Theorem 5.3.1). The Riemannian curvature tensor is denoted by $R(x)[u,v]$ where $x \in \mathcal{M}$, $u, v \in \mathcal{T}_x\mathcal{M}$ and is defined in terms of the connection $\nabla$ (Absil et al., 2009, Theorem 5.3.1). The sectional curvature $K(x)[u,v]$ for $x \in \mathcal{M}$ and $u, v \in \mathcal{T}_x\mathcal{M}$ is then defined in Lee (1997, Prop. 8.8).

$$K(x)[u,v] = \frac{\langle R(x)[u,v]u, v \rangle}{\langle u, u \rangle \langle v, v \rangle - \langle u, v \rangle^2}, \ x \in \mathcal{M}, \ u, v \in \mathcal{T}_x\mathcal{M}.$$

Denote the distance (induced by the Riemannian metric) between two points in $\mathcal{M}$ by $d(x,y)$. A geodesic $\gamma : \mathbb{R} \to \mathcal{M}$ is a constant speed curve whose length is equal to $d(x,y)$, so it is the shortest path on manifold linking $x$ and $y$. $\gamma_{x \to y}$ denotes the geodesic from $x$ to $y$ (thus $\gamma_{x \to y}(0) = x$ and $\gamma_{x \to y}(1) = y$).

The exponential map $\text{Exp}_x(v)$ maps $v \in \mathcal{T}_x\mathcal{M}$ to $y \in \mathcal{M}$ such that there exists a geodesic $\gamma$ with $\gamma(0) = x, \gamma(1) = y$ and $\frac{d}{dt}\gamma(0) = v$. The injectivity radius at point $x \in \mathcal{M}$ is the maximal radius $r$ for which the exponential map is a diffeomorphism on $\mathbb{B}_x(r) \subset \mathcal{T}_x\mathcal{M}$. The injectivity radius of the manifold, denoted by $\mathfrak{I}$, is the infimum of the injectivity radii at all points. Since the manifold is complete, we have $\mathfrak{I} > 0$. When $x, y \in \mathcal{M}$ satisfies $d(x,y) \leq \mathfrak{I}$, the exponential map admits an inverse $\text{Exp}_x^{-1}(y)$, which satisfies $d(x,y) = \|\text{Exp}_x^{-1}(y)\|$. Parallel translation $\Gamma_x^y$ denotes a the map which transports $v \in \mathcal{T}_x\mathcal{M}$ to $\Gamma_x^y v \in \mathcal{T}_y\mathcal{M}$ along $\gamma_{x \to y}$ such that the vector stays constant by satisfying a zero-acceleration condition (Lee, 1997, equation (4.13)).

For a smooth function $f : \mathcal{M} \to \mathbb{R}$, $\text{grad} f(x) \in \mathcal{T}_x\mathcal{M}$ denotes the Riemannian gradient of $f$ at $x \in \mathcal{M}$ which satisfies $\frac{d}{dt}f(\gamma(t)) = \langle \gamma'(t), \text{grad} f(x) \rangle$ (see Absil et al., 2009, Sec 3.5.1 and (3.31)). The Hessian of $f$ is defined jointly with the Riemannian structure of the manifold. The (directional) Hessian is $H(x)[\xi_x] := \nabla_{\xi_x} \text{grad} f$, and we use $H(x)[u,v] := \langle u, H(x)[v] \rangle$ as a shorthand. We call $x \in \mathcal{M}$ an $(\epsilon, -\sqrt{\rho\epsilon})$ saddle point when $\|\nabla f(x)\| \leq \epsilon$ and $\lambda_{\min}(H(x)) \leq -\sqrt{\rho\epsilon}$. We refer the interested reader to Do Carmo (2016) and Lee (1997) which provide a thorough review on these important concepts of Riemannian geometry.

## 3 Perturbed Riemannian gradient algorithm

Our main Algorithm 1 runs as follows:

1. Check the norm of the gradient: If it is large, do one step of Riemannian gradient descent, consequently the function value decreases.

2. If the norm of gradient is small, it's either an approximate saddle point or a local minimum. Perturb the variable by adding an appropriate level of noise in its tangent space, map it back to the manifold and run a few iterations.

   (a) If the function value decreases, iterates are escaping from the approximate saddle point (and the algorithm continues)

   (b) If the function value does not decrease, then it is an approximate local minimum (the algorithm terminates).

Algorithm 1 relies on the manifold's exponential map, and is useful for cases where this map is easy to compute[4]. We refer readers to Lee (1997, pp. 81-86) for the exponential map of sphere and hyperbolic manifolds, and Absil et al. (2009, Example 5.4.2, 5.4.3) for the Stiefel and Grassmann manifolds. If the exponential map is not computable, the algorithm can use a retraction[5] instead,

**Algorithm 1** Perturbed Riemannian gradient algorithm

---

**Require:** Initial point $x_0 \in \mathcal{M}$, parameters $\beta, \rho, K, \mathfrak{I}$, accuracy $\epsilon$, probability of success $\delta$ (parameters defined in Assumptions 1, 2, 3 and assumption of Theorem 1).

Set constants: $\hat{c} \geq 4$, $C := C(K, \beta, \rho)$ (defined in Lemma 2 and proof of Lemma 8)

and $\sqrt{c_{\max}} \leq \frac{1}{56\hat{c}^2}$, $r = \frac{\sqrt{c_{\max}}}{\chi^2}\epsilon$, $\chi = 3\max\{\log(\frac{d\beta(f(x_0)-f^*)}{\hat{c}\epsilon^2\delta}), 4\}$.

Set threshold values: $f_{\text{thres}} = \frac{c_{\max}}{\chi^3}\sqrt{\frac{\epsilon^3}{\rho}}$, $g_{\text{thres}} = \frac{\sqrt{c_{\max}}}{\chi^2}\epsilon$, $t_{\text{thres}} = \frac{\chi}{c_{\max}}\frac{\beta}{\sqrt{\rho\epsilon}}$, $t_{\text{noise}} = -t_{\text{thres}}-1$.

Set stepsize: $\eta = \frac{c_{\max}}{\beta}$.

**while** 1 **do**

    **if** $\|\text{grad}f(x_t)\| \leq g_{\text{thres}}$ and $t - t_{\text{noise}} > t_{\text{thres}}$ **then**

        $t_{\text{noise}} \leftarrow t, \tilde{x}_t \leftarrow x_t, x_t \leftarrow \text{Exp}_{x_t}(\xi_t), \xi_t$ uniformly sampled from $\mathbb{B}_{x_t}(r) \subset \mathcal{T}_x\mathcal{M}$.

    **end if**

    **if** $t - t_{\text{noise}} = t_{\text{thres}}$ and $f(x_t) - f(\tilde{x}_{t_{\text{noise}}}) > -f_{\text{thres}}$ **then**

        **output** $\tilde{x}_{t_{\text{noise}}}$

    **end if**

    $x_{t+1}+ \leftarrow \text{Exp}_{x_t}(-\min\{\eta, \frac{\mathfrak{I}}{\|\text{grad}f(x_t)\|}\}\text{grad}f(x_t))$.

    $t \leftarrow t + 1$.

**end while**

---

however our current analysis only covers the case of the exponential map. In Figure 1, we illustrate a function with saddle point on sphere, and plot the trajectory of Algorithm 1 when it is initialized at a saddle point.

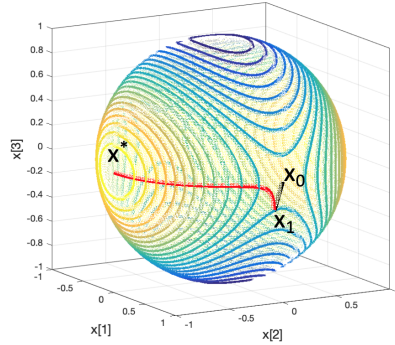

Figure 1: Function $f$ with saddle point on a sphere. $f(x) = x_1^2 - x_2^2 + 4x_3^2$. We plot the contour of this function on unit sphere. Algorithm 1 initializes at $x_0 = [1, 0, 0]$ (a saddle point), perturbs it towards $x_1$ and runs Riemannian gradient descent, and terminates at $x^* = [0, -1, 0]$ (a local minimum). We amplify the first iteration to make saddle perturbation visible.

## 4 Main theorem: escape rate for perturbed Riemannian gradient descent

We now turn to our main results, beginning with our assumptions and a statement of our main theorem. We then develop a brief proof sketch.

Our main result involves two conditions on function $f$ and one on the curvature of the manifold $\mathcal{M}$.

**Assumption 1** (Lipschitz gradient). *There is a finite constant $\beta$ such that*

$$\|\text{grad}f(y) - \Gamma_x^y \text{grad}f(x)\| \leq \beta d(x, y) \quad \textit{for all } x, y \in \mathcal{M}.$$

**Assumption 2** (Lipschitz Hessian). *There is a finite constant $\rho$ such that*

$$\|H(y) - \Gamma_x^y H(x)\Gamma_y^x\|_2 \leq \rho d(x, y) \quad \textit{for all } x, y \in \mathcal{M}.$$

**Assumption 3** (Bounded sectional curvature). *There is a finite constant $K$ such that*

$$|K(x)[u, v]| \leq K \quad \textit{for all } x \in \mathcal{M} \textit{ and } u, v \in \mathcal{T}_x\mathcal{M}$$

$K$ is an intrinsic parameter of the manifold capturing the curvature. We list a few examples here: (i) A sphere of radius $R$ has a constant sectional curvature $K = 1/R^2$ (Lee, 1997, Theorem 1.9). If

the radius is bigger, $K$ is smaller which means the sphere is less curved; (ii) A hyper-bolic space $H_R^n$ of radius $R$ has $K = -1/R^2$ (Lee, 1997, Theorem 1.9); (iii) For sectional curvature of the Stiefel and the Grasmann manifolds, we refer readers to Rapcsák (2008, Section 5) and Wong (1968), respectively.

Note that the constant $K$ is not directly related to the RLICQ parameter $R$ defined by Ge et al. (2015) which first requires describing the manifold by equality constraints. Different representations of the same manifold could lead to different curvature bounds, while sectional curvature is an intrinsic property of manifold. If the manifold is a sphere $\sum_{i=1}^{d+1} x_i^2 = R^2$, then $K = 1/R^2$, but more generally there is no simple connection. The smoothness parameters we assume are natural compared to some quantity from complicated compositions Lee et al. (2017) (Section 5.5) or pullback (Zhang & Zhang, 2018). With these assumptions, the main result of this paper is the following:

**Theorem 1.** *Under Assumptions 1,2,3, let $C(K, \beta, \rho)$ be a function defined in Lemma 2, $\hat{\rho} = \max\{\rho, C(K, \beta, \rho)\}$, if $\epsilon$ satisfies that*

$$\epsilon \leq \min \left\{ \frac{\hat{\rho}}{56 \max\{c_2(K), c_3(K)\} \eta \beta} \log\left(\frac{d\beta}{\sqrt{\hat{\rho}\epsilon}\delta}\right), \left(\frac{\Im\hat{\rho}}{12\hat{c}\sqrt{\eta\beta}} \log\left(\frac{d\beta}{\sqrt{\hat{\rho}\epsilon}\delta}\right)\right)^2 \right\} \quad (2)$$

*where $c_2(K)$, $c_3(K)$ are defined in Lemma 4, then with probability $1 - \delta$, perturbed Riemannian gradient descent with step size $c_{\max}/\beta$ converges to a $(\epsilon, -\sqrt{\hat{\rho}\epsilon})$-stationary point of $f$ in*

$$O\left(\frac{\beta(f(x_0) - f(x^*))}{\epsilon^2} \log^4\left(\frac{\beta d(f(x_0) - f(x^*))}{\epsilon^2 \delta}\right)\right)$$

*iterations.*

**Proof roadmap.** For a function satisfying smoothness condition (Assumption 1 and 2), we use a local upper bound of the objective based on the third-order Taylor expansion (see supplementary material Section A for a review),

$$f(u) \leq f(x) + \langle \mathrm{grad}f(x), \mathrm{Exp}_x^{-1}(u)\rangle + \frac{1}{2}H(x)[\mathrm{Exp}_x^{-1}(u), \mathrm{Exp}_x^{-1}(u)] + \frac{\rho}{6}\|\mathrm{Exp}_x^{-1}(u)\|^3.$$

When the norm of the gradient is large (not near a saddle), the following lemma guarantees the decrease of the objective function in one iteration.

**Lemma 1.** *(Boumal et al., 2018) Under Assumption 1, by choosing $\bar{\eta} = \min\{\eta, \frac{\Im}{\|\mathrm{grad}f(u)\|}\} = O(1/\beta)$, the Riemannian gradient descent algorithm is monotonically descending, $f(u^+) \leq f(u) - \frac{1}{2}\bar{\eta}\|\mathrm{grad}f(u)\|^2$.*

Thus our main challenge in proving the main theorem is the Riemannian gradient behaviour at an approximate saddle point:

1. Similar to the Euclidean case studied by Jin et al. (2017a), we need to bound the "thickness" of the "stuck region" where the perturbation fails. We still use a pair of hypothetical auxiliary sequences and study the "coupling" sequences. When two perturbations couple in the thinnest direction of the stuck region, their distance grows and one of them escapes from saddle point.

2. However our iterates are evolving on a manifold rather than a Euclidean space, so our strategy is to map the iterates back to an appropriate fixed tangent space where we can use the Euclidean analysis. This is done using the inverse of the exponential map and various parallel transports.

3. Several key challenges arise in doing this. Unlike Jin et al. (2017a), the structure of the manifold interacts with the local approximation of the objective function in a complicated way. On the other hand, unlike recent work on Riemannian optimization by Boumal et al. (2016a), we do not have access to a second order oracle and we need to understand how the sectional curvature and the injectivity radius (which both capture intrinsic manifold properties) affect the behavior of the first order iterates.

4. Our main contribution is to carefully investigate how the various approximation errors arising from (a) the linearization of the iteration couplings and (b) their mappings to a common tangent space can be handled on manifolds with bounded sectional curvature. We address these challenges in a sequence of lemmas (Lemmas 3 through 6) we combine to linearize the coupling iterations in a common tangent space and precisely control the approximation error. This result is formally stated in the following lemma.

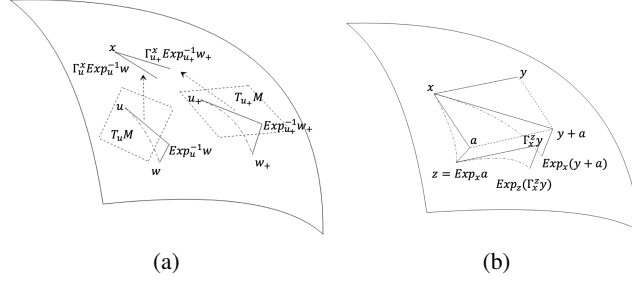

(a)                                        (b)

Figure 2: (a) Eq. (5). First map $w$ and $w_+$ to $\mathcal{T}_u\mathcal{M}$ and $\mathcal{T}_{u_+}\mathcal{M}$, and transport the two vectors to $\mathcal{T}_x\mathcal{M}$, and get their relation. (b) Lemma 3 bounds the difference of two steps starting from $x$: (1) take $y + a$ step in $\mathcal{T}_x\mathcal{M}$ and map it to manifold, and (2) take $a$ step in $\mathcal{T}_x\mathcal{M}$, map to manifold, call it $z$, and take $\Gamma_x^z y$ step in $\mathcal{T}_x\mathcal{M}$, and map to manifold. $\text{Exp}_z(\Gamma_x^z y)$ is close to $\text{Exp}_x(y + a)$.

**Lemma 2.** *Define* $\gamma = \sqrt{\tilde{\rho}\epsilon}$, $\kappa = \frac{\beta}{\gamma}$, *and* $\mathscr{S} = \sqrt{\eta\beta}\frac{\gamma}{\tilde{\rho}}\log^{-1}(\frac{d\kappa}{\delta})$. *Let us consider* $x$ *be a* $(\epsilon, -\sqrt{\tilde{\rho}\epsilon})$ *saddle point, and define* $u^+ = \text{Exp}_u(-\eta\text{grad}f(u))$ *and* $w^+ = \text{Exp}_w(-\eta\text{grad}f(w))$. *Under Assumptions 1, 2, 3, if all pairwise distances between* $u, w, u^+, w^+, x$ *are less than* $12\mathscr{S}$, *then for some explicit constant* $C(K, \rho, \beta)$ *depending only on* $K, \rho, \beta$, *there is*

$$\|\text{Exp}_x^{-1}(w^+) - \text{Exp}_x^{-1}(u^+) - (I - \eta H(x))(\text{Exp}_x^{-1}(w) - \text{Exp}_x^{-1}(u))\| \tag{3}$$
$$\leq C(K, \rho, \beta)d(u, w)\left(d(u, w) + d(u, x) + d(w, x)\right).$$

The proof of this lemma includes novel contributions by strengthen known result (Lemmas 3) and also combining known inequalities in novel ways (Lemmas 4 to 6) that allow us to control all the approximation errors and arrive at the tight rate of escape for the algorithm.

## 5 Proof of Lemma 2

Lemma 2 controls the error of the linear approximation of the iterates when mapped in $T_x\mathcal{M}$. In this section, we assume that all points are within a region of diameter $R := 12\mathscr{S} \leq \mathfrak{I}$ (inequality follows from Eq. (2) ), i.e., the distance of any two points in the following lemmas are less than $R$. The proof of Lemma 2 is based on the sequence of following lemmas.

**Lemma 3.** *Let* $x \in \mathcal{M}$ *and* $y, a \in T_x\mathcal{M}$. *Let us denote by* $z = \text{Exp}_x(a)$ *then under Assumption 3*

$$d(\text{Exp}_x(y + a), \text{Exp}_z(\Gamma_x^z y)) \leq c_1(K)\min\{\|a\|, \|y\|\}(\|a\| + \|y\|)^2. \tag{4}$$

This lemma tightens the result of Karcher (1977, C2.3), which only shows an upper-bound $O(\|a\|(\|a\| + \|y\|)^2)$. We prove the upper-bound $O(\|y\|(\|a\| + \|y\|)^2)$ in the supplement. We also need the following lemma showing that both the exponential map and its inverse are Lipschitz.

**Lemma 4.** *Let* $x, y, z \in M$, *and the distance of each two points is no bigger than* $R$. *Then under assumption 3*

$$(1 + c_2(K)R^2)^{-1}d(y, z) \leq \|\text{Exp}_x^{-1}(y) - \text{Exp}_x^{-1}(z)\| \leq (1 + c_3(K)R^2)d(y, z).$$

Intuitively this lemma relates the norm of the difference of two vectors of $\mathcal{T}_x\mathcal{M}$ to the distance between the corresponding points on the manifold $\mathcal{M}$ and follows from bounds on the Hessian of the square-distance function (Sakai, 1996, Ex. 4 p. 154). The upper-bound is directly proven by Karcher (1977, Proof of Cor. 1.6), and we prove the lower-bound via Lemma 3 in the supplement.

The following contraction result is fairly classical and is proven using the Rauch comparison theorem from differential geometry (Cheeger & Ebin, 2008).

**Lemma 5.** *(Mangoubi et al., 2018, Lemma 1) Under Assumption 3, for* $x, y \in \mathcal{M}$ *and* $w \in T_x\mathcal{M}$,

$$d(\text{Exp}_x(w), \text{Exp}_y(\Gamma_x^y w)) \leq c_4(K)d(x, y).$$

Finally we need the following corollary of the Ambrose-Singer theorem (Ambrose & Singer, 1953).

**Lemma 6.** *(Karcher, 1977, Section 6) Under Assumption 3, for* $x, y, z \in \mathcal{M}$ *and* $w \in T_x\mathcal{M}$,

$$\|\Gamma_y^z\Gamma_x^y w - \Gamma_x^z w\| \leq c_5(K)d(x, y)d(y, z)\|w\|.$$

Lemma 3 through 6 are mainly proven in the literature, and we make up the missing part in Supplementary material Section B. Then we prove Lemma 2 in Supplementary material Section B.

The spirit of the proof is to linearize the manifold using the exponential map and its inverse, and to carefully bounds the various error terms caused by the approximation. Let us denote by $\theta = d(u, w) + d(u, x) + d(w, x)$.

1. We first show using twice Lemma 3 and Lemma 5 that

$$d(\text{Exp}_u(\text{Exp}_u^{-1}(w) - \eta \Gamma_w^u \text{grad} f(w)), \text{Exp}_u(-\eta \text{grad} f(u) + \Gamma_{u_+}^u \text{Exp}_{u_+}^{-1}(w_+))) = O(\theta d(u, w)).$$

2. We use Lemma 4 to linearize this iteration in $\mathcal{T}_u \mathcal{M}$ as

$$\|\Gamma_{u_+}^u \text{Exp}_{u_+}^{-1}(w_+) - \text{Exp}_u^{-1}(w) + \eta[\text{grad} f(u) - \Gamma_w^u \text{grad} f(w)]\| = O(\theta d(u, w)).$$

3. Using the Hessian Lipschitzness

$$\|\Gamma_{u_+}^u \text{Exp}_{u_+}^{-1}(w_+)) - \text{Exp}_u^{-1}(w) + \eta H(u) \text{Exp}_u^{-1}(w)\| = O(\theta d(u, w)).$$

3. We use Lemma 6 to map to $T_x \mathcal{M}$ and the Hessian Lipschitzness to compare $H(u)$ to $H(x)$. This is an important intermediate result (see Lemma 1 in Supplementary material Section B).

$$\|\Gamma_{u_+}^x \text{Exp}_{u_+}^{-1}(w_+) - \Gamma_u^x \text{Exp}_u^{-1}(w) + \eta H(x) \Gamma_u^x \text{Exp}_u^{-1}(w)\| = O(\theta d(u, w)). \tag{5}$$

4. We use Lemma 3 and 4 to approximate two iteration updates in $\mathcal{T}_x \mathcal{M}$.

$$\|\text{Exp}_x^{-1}(w) - (\text{Exp}_x^{-1}(u) + \Gamma_u^x \text{Exp}_u^{-1}(w))\| \leq O(\theta d(u, w)). \tag{6}$$

And same for the $u_+, w_+$ pair replacing $u, w$.

5. Combining Eq. (5) and Eq. (6) together, we obtain

$$\|\text{Exp}_x^{-1}(w^+) - \text{Exp}_x^{-1}(u^+) - (I - \eta H(x))(\text{Exp}_x^{-1}(w) - \text{Exp}_x^{-1}(u))\| \leq O(\theta d(u, w)).$$

Now note that, the iterations $u, u_+, w, w_+$ of the algorithm are both on the manifold. We use $\text{Exp}_x^{-1}(\cdot)$ to map them to the same tangent space at $x$.

Therefore we have linearized the two coupled trajectories $\text{Exp}_x^{-1}(u_t)$ and $\text{Exp}_x^{-1}(w_t)$ in a common tangent space, and we can modify the Euclidean escaping saddle analysis thanks to the error bound we proved in Lemma 2.

## 6 Proof of main theorem

In this section we suppose all assumptions in Section 4 hold. The proof strategy is to show with high probability that the function value decreases of $\mathscr{F}$ in $\mathscr{T}$ iterations at an approximate saddle point. Lemma 7 suggests that, if after a perturbation and $\mathscr{T}$ steps, the iterate is $\Omega(\mathscr{S})$ far from the approximate saddle point, then the function value decreases. If the iterates do not move far, the perturbation falls in a stuck region. Lemma 8 uses a coupling strategy, and suggests that the width of the stuck region is small in the negative eigenvector direction of the Riemannian Hessian.

Define

$$\mathscr{F} = \eta \beta \frac{\gamma^3}{\hat{\rho}^2} \log^{-3}(\frac{d\kappa}{\delta}), \ \mathscr{G} = \sqrt{\eta \beta} \frac{\gamma^2}{\hat{\rho}} \log^{-2}(\frac{d\kappa}{\delta}), \ \mathscr{T} = \frac{\log(\frac{d\kappa}{\delta})}{\eta \gamma}.$$

At an approximate saddle point $\tilde{x}$, let $y$ be in the neighborhood of $\tilde{x}$ where $d(y, \tilde{x}) \leq \Im$, denote

$$\tilde{f}_y(x) := f(y) + \langle \text{grad} f(y), \text{Exp}_y^{-1}(\tilde{x}) \rangle + \frac{1}{2} \Gamma_{\tilde{x}}^y H(\tilde{x}) \Gamma_y^{\tilde{x}} [\text{Exp}_y^{-1}(\tilde{x}), \text{Exp}_y^{-1}(\tilde{x})].$$

Let $\|\text{grad} f(\tilde{x})\| \leq \mathscr{G}$ and $\lambda_{\min}(H(\tilde{x})) \leq -\gamma$. We consider two iterate sequences, $u_0, u_1, ...$ and $w_0, w_1, ...$ where $u_0, w_0$ are two perturbations at $\tilde{x}$.

**Lemma 7.** *Assume Assumptions 1, 2, 3 and Eq. (2) hold. There exists a constant $c_{\max}$, $\forall \hat{c} > 3, \delta \in (0, \frac{d\kappa}{e}]$, for any $u_0$ with $d(\tilde{x}, u_0) \leq 2\mathscr{S}/(\kappa \log(\frac{d\kappa}{\delta}))$, $\kappa = \beta/\gamma$.*

$$T = \min \left\{ \inf_t \left\{ t | \tilde{f}_{u_0}(u_t) - f(u_0) \leq -3\mathscr{F} \right\}, \hat{c}\mathscr{T} \right\},$$

*then $\forall \eta \leq c_{\max}/\beta$, we have $\forall 0 < t < T$, $d(u_0, u_t) \leq 3(\hat{c}\mathscr{S})$.*

**Lemma 8.** *Assume Assumptions 1, 2, 3 and Eq. (2) hold. Take two points $u_0$ and $w_0$ which are perturbed from an approximate saddle point, where $d(\tilde{x}, u_0) \leq 2\mathscr{S}/(\kappa \log(\frac{d\kappa}{\delta}))$, $\mathrm{Exp}_{\tilde{x}}^{-1}(w_0) - \mathrm{Exp}_{\tilde{x}}^{-1}(u_0) = \mu r e_1$, $e_1$ is the smallest eigenvector[6] of $H(\tilde{x})$, $\mu \in [\delta/(2\sqrt{d}), 1]$, and the algorithm runs two sequences $\{u_t\}$ and $\{w_t\}$ starting from $u_0$ and $w_0$. Denote*

$$T = \min\left\{\inf_t \left\{t | \tilde{f}_{w_0}(w_t) - f(w_0) \leq -3\mathscr{F}\right\}, \hat{c}\mathscr{T}\right\},$$

*then $\forall \eta \leq c_{\max}/l$, if $\forall 0 < t < T$, $d(\tilde{x}, u_t) \leq 3(\hat{c}\mathscr{S})$, we have $T < \hat{c}\mathscr{T}$.*

We prove Lemma 7 and 8 in supplementary material Section C. We also prove, in the same section, the main theorem using the coupling strategy of Jin et al. (2017a). but with the additional difficulty of taking into consideration the effect of the Riemannian geometry (Lemma 2) and the injectivity radius.

## 7 Examples

**kPCA.** We consider the kPCA problem, where we want to find the $k \leq n$ principal eigenvectors of a symmetric matrix $H \in \mathbb{R}^{n \times n}$, as an example (Tripuraneni et al., 2018). This corresponds to

$$\min_{X \in \mathbb{R}^{n \times k}} -\frac{1}{2}\mathrm{tr}(X^T H X) \quad \text{subject to } X^T X = I,$$

which is an optimization problem on the Grassmann manifold defined by the constraint $X^T X = I$. If the eigenvalues of $H$ are distinct, we denote by $v_1,...,v_n$ the eigenvectors of $H$, corresponding to eigenvalues with decreasing order. Let $V^* = [v_1, ..., v_k]$ be the matrix with columns composed of the top $k$ eigenvectors of $H$, then the local minimizers of the objective function are $V^*G$ for all unitary matrices $G \in \mathbb{R}^{k \times k}$. Denote also by $V = [v_{i_1}, ..., v_{i_k}]$ the matrix with columns composed of $k$ distinct eigenvectors, then the first order stationary points of the objective function (with Riemannian gradient being 0) are $VG$ for all unitary matrices $G \in \mathbb{R}^{k \times k}$. In our numerical experiment, we choose $H$ to be a diagonal matrix $H = \mathrm{diag}(0, 1, 2, 3, 4)$ and let $k = 3$. The Euclidean basis $(e_i)$ are an eigenbasis of $H$ and the first order stationary points of the objective function are $[e_{i_1}, e_{i_2}, e_{i_3}]G$ with distinct basis and $G$ being unitary. The local minimizers are $[e_3, e_4, e_5]G$. We start the iteration at $X_0 = [e_2, e_3, e_4]$ and see in Fig. 3 the algorithm converges to a local minimum.

**Burer-Monteiro approach for certain low rank problems.** Following Boumal et al. (2016b), we consider, for $A \in \mathbb{S}^{d \times d}$ and $r(r + 1)/2 \leq d$, the problem

$$\min_{X \in \mathbb{S}^{d \times d}} \mathrm{trace}(AX), \ s.t. \ \mathrm{diag}(X) = 1, X \succeq 0, \mathrm{rank}(X) \leq r.$$

We factorize $X$ by $YY^T$ with an overparametrized $Y \in \mathbb{R}^{d \times p}$ and $p(p + 1)/2 \geq d$. Then any local minimum of

$$\min_{Y \in \mathbb{R}^{d \times p}} \mathrm{trace}(AYY^T), \ s.t. \ \mathrm{diag}(YY^T) = 1,$$

is a global minimum where $YY^T = X^*$ (Boumal et al., 2016b). Let $f(Y) = \frac{1}{2}\mathrm{trace}(AYY^T)$. In the experiment, we take $A \in \mathbb{R}^{100 \times 20}$ being a sparse matrix that only the upper left $5 \times 5$ block is random and other entries are 0. Let the initial point $Y_0 \in \mathbb{R}^{100 \times 20}$, such that $(Y_0)_{i,j} = 1$ for $5j - 4 \leq i \leq 5j$ and $(Y_0)_{i,j} = 0$ otherwise. Then $Y_0$ is a saddle point. We see in Fig. 3 the algorithm converges to the global optimum.

**Summary** We have shown that for the constrained optimization problem of minimizing $f(x)$ subject to a manifold constraint as long as the function and the manifold are appropriately smooth, a perturbed Riemannian gradient descent algorithm will escape saddle points with a rate of order $1/\epsilon^2$ in the accuracy $\epsilon$, polylog in manifold dimension $d$, and depends polynomially on the curvature and smoothness parameters.

A natural extension of our result is to consider other variants of gradient descent, such as the heavy ball method, Nesterov's acceleration, and the stochastic setting. The question is whether these algorithms with appropriate modification (with manifold constraints) would have a fast convergence to second-order stationary point (not just first-order stationary as studied in recent literature), and whether it is possible to show the relationship between convergence rate and smoothness of manifold.

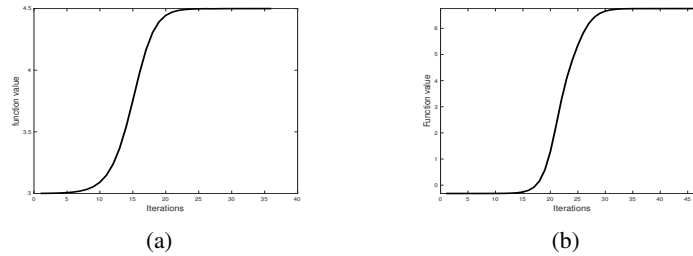

(a)                                  (b)

Figure 3: (a) kPCA problem. We start from an approximate saddle point, and it converges to a local minimum (which is also global minimum). (b) Burer-Monteiro approach Plot $f(Y) = \frac{1}{2}\text{trace}(AYY^T)$ versus iterations. We start from the saddle point, and it converges to a local minimum (which is also global minimum).

## Footnotes

[1]Here $d$ is the dimension of the manifold itself; we do not consider $\mathcal{M}$ as a submanifold of a higher dimensional space. For instance, if $\mathcal{M}$ is a 2-dimensional sphere embedded in $\mathbb{R}^3$, its dimension is $d = 2$.

[2]defined as $x$ satisfying $\|\nabla f(x)\| \leq \epsilon$, $\lambda_{\min}\nabla^2 f(x) \geq -\sqrt{\rho\epsilon}$

[3]Since our results are local, completeness is not necessary and our results can be easily generalized, with extra assumptions on the injectivity radius.

[4]Numerous interesting manifolds have closed-form exponential maps: *the Grassmannian manifold, the Stiefel manifold, the Minkowski space, the hyperbolic space, $SE(n)$, $SO(n)$...*(see, Miolane et al. (2018); Boumal et al. (2014) and their open-source packages and Bécigneul & Ganea (2019, Sec 5) for an example in NLP).

[5]A retraction is a first-order approximation of the exponential map which is often easier to compute.

[6]"smallest eigenvector" means the eigenvector corresponding to the smallest eigenvalue.

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
