[Supplementary Material · supplementary.pdf]

## 1 Appendix

### 2 Organization of the Appendix

3 In Appendix A we review classical results on the Taylor expansions for functions on Riemannian
4 manifold. In Appendix B we provide the proof of Lemma 2 which requires to expand the iterates on
5 the tangent space in the the saddle point. Finally, in Appendix C, we provide the proofs of Lemma 7
6 and Lemma 8 which enable to prove the main theorem of the paper.

7 Throughout the paper we assume that the objective function and the manifold are smooth. Here we
8 list the assumptions that are used in the following lemmas.

9 **Assumption 1** (Lipschitz gradient)**.** *There is a finite constant $\beta$ such that*

$$\|\mathrm{grad}f(y) - \Gamma_x^y\mathrm{grad}f(x)\| \leq \beta d(x,y) \quad \textit{for all } x, y \in \mathcal{M}.$$

10 **Assumption 2** (Lipschitz Hessian)**.** *There is a finite constant $\rho$ such that*

$$\|H(y) - \Gamma_x^y H(x)\Gamma_y^x\|_2 \leq \rho d(x,y) \quad \textit{for all } x, y \in \mathcal{M}.$$

11 **Assumption 3** (Bounded sectional curvature)**.** *There is a finite constant $K$ such that*

$$|K(x)[u,v]| \leq K \quad \textit{for all } x \in \mathcal{M} \textit{ and } u, v \in \mathcal{T}_x\mathcal{M}$$

## 12 A Taylor expansions on Riemannian manifold

13 We provide here the Taylor expansion for functions and gradients of functions defined on a Riemannian
14 manifold.

### 15 A.1 Taylor expansion for the gradient

16 For any point $x \in \mathcal{M}$ and $z \in \mathcal{M}$ be a point in the neighborhood of $x$ where the geodesic $\gamma_{x \to z}$ is
17 defined.

$$\begin{aligned}
\Gamma_z^x(\mathrm{grad}f(z)) &= \mathrm{grad}f(x) + \nabla_{\gamma'_{x \to z}(0)}\mathrm{grad}f + \int_0^1 (\Gamma_{\gamma_{x \to z}(\tau)}^x \nabla_{\gamma'_{x \to z}(\tau)}\mathrm{grad}f - \nabla_{\gamma'_{x \to z}(0)}\mathrm{grad}f)dx_\tau \\
&= \mathrm{grad}f(x) + \nabla_{\gamma'_{x \to z}(0)}\mathrm{grad}f + \Delta(z),
\end{aligned} \tag{1}$$

18 where $\Delta(z) := \int_0^1 (\Gamma_{\gamma_{x \to z}(\tau)}^x \nabla_{\gamma'_{x \to z}(\tau)}\mathrm{grad}f - \nabla_{\gamma'_{x \to z}(0)}\mathrm{grad}f)d\tau$. The Taylor approximation in
19 Eq. (1) is proven by Absil et al. (2009, Lemma 7.4.7).

### 20 A.2 Taylor expansion for the function

21 Taylor expansion of the gradient enables us to approximate the iterations of the main algorithm, but
22 obtaining the convergence rate of the algorithm requires proving that the function value decreases
23 following the iterations. We need to give the Taylor expansion of $f$ with the parallel translated
24 gradient on LHS of Eq. (1). To simplify the notation, let $\gamma$ denote the $\gamma_{x \to z}$.

$$f(z) - f(x) = \int_0^1 \frac{d}{d\tau}f(\gamma(\tau))d\tau \tag{2a}$$

$$= \int_0^1 \langle \gamma'(\tau), \mathrm{grad}f(\gamma(\tau)) \rangle d\tau \tag{2b}$$

$$= \int_0^1 \langle \Gamma_{\gamma(\tau)}^x \gamma'(\tau), \Gamma_{\gamma(\tau)}^x \mathrm{grad}f(\gamma(\tau)) \rangle d\tau \tag{2c}$$

$$= \int_0^1 \langle \gamma'(0), \Gamma_{\gamma(\tau)}^0 \mathrm{grad}f(\gamma(\tau)) \rangle d\tau \tag{2d}$$

$$= \int_0^1 \langle \gamma'(0), \mathrm{grad}f(x) + \nabla_{\tau\gamma'(0)}\mathrm{grad}f + \Delta(\gamma(\tau)) \rangle d\tau \tag{2e}$$

$$= \langle \gamma'(0), \mathrm{grad}f(x) + \tfrac{1}{2}\nabla_{\gamma'(0)}\mathrm{grad}f + \bar{\Delta}(z) \rangle. \tag{2f}$$

Figure 1: Lemma 1. First map $w$ and $w_+$ to $\mathcal{T}_u\mathcal{M}$ and $\mathcal{T}_{u_+}\mathcal{M}$, and transport the two vectors to $\mathcal{T}_x\mathcal{M}$, and get their relation.

$\Delta(z)$ is defined in Eq. (1). $\bar{\Delta}(z) = \int_0^1 \Delta(\gamma(\tau))d\tau$. The second line is just rewriting by definition. Eq. (2c) means the parallel translation preserves the inner product (Tu, 2017, Prop. 14.16). Eq. (2d) uses $\Gamma^x_{\gamma(t)}\gamma'(t) = \gamma'(0)$, meaning that the velocity stays constant along a geodesic (Absil et al., 2009, (5.23)). Eq. (2e) uses Eq. (1). In Euclidean space, the Taylor expansion is

$$f(z) - f(x) = \langle z, \nabla f(x) + \nabla^2 f(x)z + \int_0^1 (\nabla^2 f(\tau z) - \nabla^2 f(x))z d\tau \rangle. \tag{3}$$

Compare Eq. (2) and Eq. (3), $z$ is replaced by $\gamma'(0) := \gamma'_{x \to z}(0)$ and $\tau z$ is replaced by $\tau\gamma'_{x \to z}(0)$ or $\gamma_{x \to z}(\tau)$.

Now we have

$$f(u_t) = f(x) + \langle \gamma'(0), \text{grad} f(x) \rangle + \frac{1}{2}H(x)[\gamma'(0), \gamma'(0)] + \langle \gamma'(0), \bar{\Delta}(u_t) \rangle.$$

## B  Linearization of the iterates in a fixed tangent space

In this section we linearize the progress of the iterates of our algorithm in a fixed tangent space $\mathcal{T}_x\mathcal{M}$. We always assume here that all points are within a region of diameter $R := 12\mathscr{S} \leq \mathfrak{I}$. In the course of the proof we need several auxilliary lemmas which are stated in the last two subsections of this section.

### B.1  Evolution of $\text{Exp}_u^{-1}(w)$

We first consider the evolution of $\text{Exp}_u^{-1}(w)$ in a fixed tangent space $\mathcal{T}_x\mathcal{M}$. We show in the following lemma that it approximately follows a linear reccursion.

**Lemma 1.** *Define* $\gamma = \sqrt{\bar{\rho}\epsilon}$, $\kappa = \frac{\beta}{\gamma}$, *and* $\mathscr{S} = \sqrt{\eta\beta}\frac{\gamma}{\bar{\rho}}\log^{-1}(\frac{d\kappa}{\delta})$. *Let us consider* $x$ *be a* $(\epsilon, -\sqrt{\bar{\rho}\epsilon})$ *saddle point, and define* $u^+ = \text{Exp}_u(-\eta\text{grad} f(u))$ *and* $w^+ = \text{Exp}_w(-\eta\text{grad} f(w))$. *Under Assumptions 1, 2, 3, if all pairwise distances between* $u, w, u^+, w^+, x$ *are less than* $12\mathscr{S}$, *then for some explicit constant* $C_1(K, \rho, \beta)$ *depending only on* $K, \rho, \beta$, *there is*

$$\|\Gamma^x_{u^+}\text{Exp}_{u^+}^{-1}(w^+) - (I - \eta H(x))\Gamma^x_u\text{Exp}_u^{-1}(w)\|$$
$$\leq C_1(K, \rho, \beta)d(u,w)\left(d(u,w) + d(u,x) + d(w,x)\right).$$

*for some explicit function* $C_1$.

This lemma is illustrated in Fig. 1.

*Proof.* Denote $-\eta\text{grad} f(u) = v_u$, $-\eta\text{grad} f(w) = v_w$. $v$ is a smooth map. We first prove the following claim.

**Claim 1.**
$$d(u_+, w_+) \le c_6(K)d(u, w),$$
where $c_6(K) = c_4(K) + 1 + c_2(K)R^2$.

To show this, note that
$$d(u_+, w_+) \le d(u_+, \tilde{w}_+) + d(\tilde{w}_+, w_+),$$
and using Lemma 5 with $\tilde{w}_+ = \mathrm{Exp}_w(\Gamma_u^w v_u)$,
$$\begin{aligned}
d(\tilde{w}_+, w_+) &= d(\mathrm{Exp}_w(v_w), \mathrm{Exp}_w(\Gamma_u^w v_u)) \\
&\le (1 + c_2(K)R^2)\|v_w - \Gamma_u^w v_u\| \\
&\le \beta(1 + c_2(K)R^2)d(u, w).
\end{aligned}$$

Using Lemma 5,
$$d(\tilde{w}_+, u_+) \le c_4(K)d(u, w). \tag{4}$$

Adding the two inequalities proves the claim.

We use now Lemma 3 between $(u, w, u_+, w_+)$ in two different ways. First let us use it for $a = \mathrm{Exp}_u^{-1}(w)$ and $y = \Gamma_w^u v_w$. We obtain:
$$d(w_+, \mathrm{Exp}_u(\mathrm{Exp}_u^{-1}(w) + \Gamma_w^u v_w)) \le c_1(K)d(u, w)(d(u, w)^2 + \|v_w\|^2). \tag{5}$$

Then we use it for $a = \mathrm{Exp}_u^{-1}(v_u)$ and $y = \Gamma_{u_+}^u \mathrm{Exp}_{u_+}^{-1}(w_+)$ which yields
$$\begin{aligned}
&d(w_+, \mathrm{Exp}_u(v_u + \Gamma_{u_+}^u \mathrm{Exp}_{u_+}^{-1}(w_+))) \\
&\le c_1(K)d(u_+, w_+)(d(u_+, w_+)^2 + \|v_u\|^2) \\
&\le c_1(K)c_5(K, \|v_u\|, \|v_w\|)d(u, w) \cdot \left[ c_5(K, \|v_u\|, \|v_w\|)^2 d(u, w)^2 + \|v_u\|^2 \right].
\end{aligned}$$

Using the triangular inequality we have
$$\begin{aligned}
&d(\mathrm{Exp}_u(\mathrm{Exp}_u^{-1}(w) + \Gamma_w^u v_w), \mathrm{Exp}_u(v_u + \Gamma_{u_+}^u \mathrm{Exp}_{u_+}^{-1}(w_+))) \\
&\le d(w_+, \mathrm{Exp}_u(\mathrm{Exp}_u^{-1}(w) + \Gamma_w^u v_w)) + d(w_+, \mathrm{Exp}_u(v_u + \Gamma_{u_+}^u \mathrm{Exp}_{u_+}^{-1}(w_+))) \\
&\le c_7 d(u, w)
\end{aligned}$$
with $c_7$ defined as
$$c_7 = c_1(K)c_6(K) \cdot [c_5(K, \|v_u\|, \|v_w\|)^2 d(u, w)^2 + \|v_u\|^2 + \|v_w\|^2].$$

We use again Lemma 4,
$$\|\Gamma_{u_+}^u \mathrm{Exp}_{u_+}^{-1}(w_+)) - \mathrm{Exp}_u^{-1}(w) - [v_u - \Gamma_w^u v_w]\| \le (1 + c_3(K)R^2) \cdot c_7 d(u, w).$$

Therefore we have linearized the iterate in $T_u\mathcal{M}$. We should see how to transport it back to $T_x\mathcal{M}$.
With Lemma 6 we have
$$\|[\Gamma_u^x \Gamma_{u_+}^u - \Gamma_{u_+}^x]\mathrm{Exp}_{u_+}^{-1}(w_+))\| = c_5(K)d(u, x)d(u_+, w_+)\|v_u\|.$$

Note $v_u$ and $v_w$ are $-\eta \mathrm{grad} f(u)$ and $-\eta \mathrm{grad} f(w)$, we define $\nabla v(x)$ the gradient of $v$, i.e., $-\eta H$.
Using Hessian Lipschitz,
$$\begin{aligned}
&\|v_u - \Gamma_w^u v_w + \eta H(u)\mathrm{Exp}_u^{-1}(w)\| \\
&= \|v_u - \Gamma_w^u v_w - \nabla v(u)\mathrm{Exp}_u^{-1}(w)\| \\
&\le \rho d(u, w)^2,
\end{aligned}$$
and
$$\|\nabla v(u)\mathrm{Exp}_u^{-1}(w) - \Gamma_x^u \nabla v(x)\Gamma_u^x \mathrm{Exp}_u^{-1}(w)\| \le \rho d(u, w)d(u, x).$$

So we have
$$\begin{aligned}
&\|\Gamma_{u_+}^x \mathrm{Exp}_{u_+}^{-1}(w_+) - (I + \nabla v(x))\Gamma_u^x \mathrm{Exp}_u^{-1}(w)\| \\
&\le c_7 d(u, w) + \rho d(u, w)(d(u, w) + d(u, x)) + c_5(K)d(u, x)d(u_+, w_+)\|v_u\| := D_1 \tag{6}
\end{aligned}$$

$\square$

## B.2 Evolution of $\mathrm{Exp}_x^{-1}(w) - \mathrm{Exp}_x^{-1}(u)$

We consider now the evolution of $\mathrm{Exp}_x^{-1}(w) - \mathrm{Exp}_x^{-1}(u)$ in the fixed tangent space $\mathcal{T}_x\mathcal{M}$. We show in the following lemma that it also approximately follows a linear iteration.

**Lemma 2.** *Define $\gamma = \sqrt{\hat{\rho}\epsilon}$, $\kappa = \frac{\beta}{\gamma}$, and $\mathscr{S} = \sqrt{\eta\beta}\frac{\gamma}{\hat{\rho}}\log^{-1}(\frac{d\kappa}{\delta})$. Let us consider $x$ be a $(\epsilon, -\sqrt{\hat{\rho}\epsilon})$ saddle point, and define $u^+ = \mathrm{Exp}_u(-\eta\mathrm{grad}f(u))$ and $w^+ = \mathrm{Exp}_w(-\eta\mathrm{grad}f(w))$. Under Assumptions 1, 2, 3, if all pairwise distances between $u, w, u^+, w^+, x$ are less than $12\mathscr{S}$, then for some explicit constant $C(K, \rho, \beta)$ depending only on $K, \rho, \beta$, there is*

$$\|\mathrm{Exp}_x^{-1}(w^+) - \mathrm{Exp}_x^{-1}(u^+) - (I - \eta H(x))(\mathrm{Exp}_x^{-1}(w) - \mathrm{Exp}_x^{-1}(u))\| \tag{7}$$
$$\leq C(K, \rho, \beta)d(u, w)\left(d(u, w) + d(u, x) + d(w, x)\right).$$

This lemma controls the error of the linear approximation of the iterates hen mapped in $\mathcal{T}_x\mathcal{M}$ and largely follows from Lemma 1.

*Proof.* We have that

$$w = \mathrm{Exp}_x(\mathrm{Exp}_x^{-1}(w)) \tag{8}$$
$$= \mathrm{Exp}_u(\mathrm{Exp}_u^{-1}(w)). \tag{9}$$

Use Eq. (9), let $a = \mathrm{Exp}_x^{-1}(u)$ and $v = \Gamma_u^x\mathrm{Exp}_u^{-1}(w)$, Lemma 3 suggests that

$$d(\mathrm{Exp}_u(\mathrm{Exp}_u^{-1}(w)), \mathrm{Exp}_x(\mathrm{Exp}_x^{-1}(u) + \Gamma_u^x\mathrm{Exp}_u^{-1}(w)))$$
$$\leq c_1(K)\|\mathrm{Exp}_u^{-1}(w)\|(\|\mathrm{Exp}_u^{-1}(w)\| + \|\mathrm{Exp}_x^{-1}(u)\|)^2.$$

Compare with Eq. (8), we have

$$d(\mathrm{Exp}_x(\mathrm{Exp}_x^{-1}(w)), \mathrm{Exp}_x(\mathrm{Exp}_x^{-1}(u) + \Gamma_u^x\mathrm{Exp}_u^{-1}(w)))$$
$$\leq c_1(K)\|\mathrm{Exp}_u^{-1}(w)\|(\|\mathrm{Exp}_u^{-1}(w)\| + \|\mathrm{Exp}_x^{-1}(u)\|)^2$$
$$:= D. \tag{10}$$

Denote the quantity above by $D$. Now use Lemma 4

$$\|\mathrm{Exp}_x^{-1}(w) - (\mathrm{Exp}_x^{-1}(u) + \Gamma_u^x\mathrm{Exp}_u^{-1}(w))\| \leq (1 + c_3(K)R^2)D.$$

Analogously

$$\|\mathrm{Exp}_x^{-1}(w_+) - (\mathrm{Exp}_x^{-1}(u_+) + \Gamma_{u_+}^x\mathrm{Exp}_{u_+}^{-1}(w_+))\| \leq (1 + c_3(K)R^2)D_+$$

where

$$D_+ = c_1(K)\|\mathrm{Exp}_{u_+}^{-1}(w_+)\|(\|\mathrm{Exp}_{u_+}^{-1}(w_+)\| + \|\mathrm{Exp}_x^{-1}(u_+)\|)^2 \tag{11}$$

And we can compare $\Gamma_u^x\mathrm{Exp}_u^{-1}(w)$ and $\Gamma_{u_+}^x\mathrm{Exp}_{u_+}^{-1}(w_+)$ using Eq. (6). In the end we have

$$\|\mathrm{Exp}_x^{-1}(w^+) - \mathrm{Exp}_x^{-1}(u^+) - (I - \eta H(x))(\mathrm{Exp}_x^{-1}(w) - \mathrm{Exp}_x^{-1}(u))\|$$
$$\leq \|\mathrm{Exp}_x^{-1}(w_+) - (\mathrm{Exp}_x^{-1}(u_+) + \Gamma_{u_+}^x\mathrm{Exp}_{u_+}^{-1}(w_+))\|$$
$$+ \|\mathrm{Exp}_x^{-1}(w) - (\mathrm{Exp}_x^{-1}(u) + \Gamma_u^x\mathrm{Exp}_u^{-1}(w))\|$$
$$+ \|\Gamma_{u_+}^x\mathrm{Exp}_{u_+}^{-1}(w_+) - \Gamma_u^x\mathrm{Exp}_u^{-1}(w) - \nabla v(x)\Gamma_u^x\mathrm{Exp}_u^{-1}(w)\|$$
$$+ \|\nabla v(x)(\Gamma_u^x\mathrm{Exp}_u^{-1}(w) - (\mathrm{Exp}_x^{-1}(w) - \mathrm{Exp}_x^{-1}(u)))\|$$
$$\leq (1 + c_3(K)R^2)(D_+ + D) + D_1 + \eta\|H(x)\|D.$$

$D, D_+$ and $D_1$ are defined in Eq. (10), Eq. (11) and Eq. (6), they are all order $d(u, w)\left(d(u, w) + d(u, x) + d(w, x)\right)$ so we get the correct order in Eq. (7). $\qquad\square$

Figure 2: Lemma 3 bounds the difference of two steps starting from $x$: (1) take $y + a$ step in $\mathcal{T}_x\mathcal{M}$ and map it to manifold, and (2) take $a$ step in $\mathcal{T}_x\mathcal{M}$, map to manifold, call it $z$, and take $\Gamma_x^z y$ step in $\mathcal{T}_x\mathcal{M}$, and map to manifold. $\mathrm{Exp}_z(\Gamma_x^z y)$ is close to $\mathrm{Exp}_x(y + a)$.

## B.3 Control of two-steps iteration

In the following lemma we control the distance between the point obtained after moving along the sum of two vectors in the tangent space, and the point obtained after moving a first time along the first vector and then a second time along the transport of the second vector. This is illustrated in Fig. 2.

**Lemma 3.** *Let $x \in \mathcal{M}$ and $y, a \in T_x\mathcal{M}$. Let us denote by $z = \mathrm{Exp}_x(a)$ then under Assumption 3*

$$d(\mathrm{Exp}_x(y + a), \mathrm{Exp}_z(\Gamma_x^z y)) \leq c_1(K) \min\{\|a\|, \|y\|\}(\|a\| + \|y\|)^2. \tag{12}$$

This lemma which is crucial in the proofs of Lemma 2 and Lemma 1 tightens the result of Karcher (1977, C2.3), which only shows an upper-bound $O(\|a\|(\|a\| + \|y\|)^2)$.

*Proof.* We adapt the proof of Karcher (1977, Eq. (C2.3) in App C2.2), the only difference being that we bound more carefully the initial normal component. We restate here the whole proof for completeness.

Let $x \in \mathcal{M}$ and $y, a \in T_x\mathcal{M}$. We denote by $\gamma(t) = \mathrm{Exp}_x(ta)$. We want to compare the point $\mathrm{Exp}_x(r(y + a))$ and $\mathrm{Exp}_\gamma(1)(\Gamma_x^{\gamma(1)y})$. These two points , for a fixed $r$ are joined by the curve

$$t \mapsto c(r, t) = \mathrm{Exp}_{\gamma(t)}(r\Gamma_x^{\gamma(t)}(y + (1 - t)a)).$$

We note that $\frac{d}{dt}c(r, t)$ is a Jacobi field along the geodesic $r \mapsto c(r, t)$, which we denote by $J_t(r)$. We importantly remark that the length of the geodesic $r \mapsto c(r, t)$ is bounded as $\|\frac{d}{dr}c(r, t)\| \leq \|y + (1 - t)a\|$. We denote this quantity by $\rho_t = \|y + (1 - t)a\|$. The initial condition of the Jacobi field $J_t$ are given by:

$$J_t(0) = \frac{d}{dt}\gamma(t) = \Gamma_x^{\gamma(t)}a$$

$$\frac{D}{dr}J_t(0) = \frac{D}{dr}\Gamma_x^{\gamma(t)}(y + (1 - t)a) = -\Gamma_x^{\gamma(t)}a.$$

These two vectors are linearly dependent and it is therefore possible to apply Karcher (1977, Proposition A6) to bound $J_t^{\mathrm{norm}}$. Moreover, following Karcher (1977, App A0.3 ), the tangential component of the Jacobi field is known explicitly, independent of the metric, by

$$J_t^{\tan}(r) = \left(J_t^{\tan}(0) + r\frac{D}{dr}J_t^{\tan}(0)\right)\frac{d}{dr}c(r, t)$$

where the initial conditions of the tangential component of the Jacobi fields are given by $J_t^{\tan}(0) = \langle J_t(0), \frac{\frac{d}{dr}c(r,t)}{\|\frac{d}{dr}c(r,t)\|}\rangle$ and $\frac{D}{dr}J_t^{\tan}(0) = \langle \frac{D}{dr}J_t(0), \frac{\frac{d}{dr}c(r,t)}{\|\frac{d}{dr}c(r,t)\|}\rangle = -J_t^{\tan}(0)$. Therefore

$$J_t^{\tan}(r) = (1 - r)J_t^{\tan}(0)\frac{d}{dr}c(r, t),$$

105 and $J_t^{\tan}(1) = 0$.

106 We estimate now the distance $d(\mathrm{Exp}_x(y+a), \mathrm{Exp}_z(\Gamma_x^z y))$ by the length of the curve $t \mapsto c(r,t)$ as
107 follows:

$$d(\mathrm{Exp}_x(y+a), \mathrm{Exp}_z(\Gamma_x^z y)) \leq \int_0^1 \|\frac{d}{dt} c(1,t)\| dt = \int_0^1 \|J_t^{\mathrm{norm}}(1)\| dt,$$

108 where we use crucially that $J_t^{tan}(1) = 0$.

109 We utilize (Karcher, 1977, Proposition A.6) to bound $\|J_t^{\mathrm{norm}}(1)\|$ as

$$\|J_t^{\mathrm{norm}}(1)\| \leq \|J_t^{\mathrm{norm}}(0)\| (\cosh(\sqrt{K}\rho_t) - \frac{\sinh(\sqrt{K}\rho_t)}{\sqrt{K}\rho_t})$$

110 using (Karcher, 1977, Equation (A6.3)) with $\kappa = 0$, $f_\kappa(1) = 0$ and recalling that the geodesics
$r \mapsto c(r,t)$ have length $\rho_t$.

Figure 3: Figure for Lemma 3.

111

112 In particular for small value $\|a\| + \|y\|$ we have for some constant $c_1(K)$,

$$\|J_t^{\mathrm{norm}}(1)\| \leq \|J_t^{\mathrm{norm}}(0)\| c_1(K) \rho_t^2.$$

113 We bound $\|J_t^{\mathrm{norm}}(0)\|$ now. This is the main difference with the original proof of Karcher (1977)
114 who directly bounded $\|J_t^{\mathrm{norm}}(0)\| \leq \|J_t(0)\| = \|a\|$ and $\rho_t \leq \|a\| + \|y\|$. Therefore his proof does
115 not lead to the correct dependence in $\|y\|$.

116 We have $J_t^0 = \Gamma_x^{\gamma(t)} a$, and the tangential component (velocity of $r \to c(r,t)$) is in the $\Gamma_x^{\gamma(t)}(y + (1 -$
117 $t)a)$ direction. Let $\tilde{z} = \Gamma_x^{\gamma(t)}(y + (1-t)a)$ and $\mathcal{P}_{\tilde{z}^\perp}$ and $\mathcal{P}_{a^\perp}$ denote the projection onto orthogonal
118 complement of $\tilde{z}$ and $a$.

$$
\begin{aligned}
\|J_t^{\mathrm{norm}}(0)\|^2 &= \|\mathcal{P}_{\tilde{z}^\perp}(a)\|^2 \\
&= \|a\|^2 - \frac{(a^T \tilde{z})^2}{\|\tilde{z}\|^2} \\
&= \frac{\|a\|^2}{\|\tilde{z}\|^2} \left( \|\tilde{z}\|^2 - \frac{(a^T \tilde{z})^2}{\|\tilde{z}\|^2} \right) \\
&\leq \frac{\|a\|^2}{\|\tilde{z}\|^2} \|\mathcal{P}_{a^\perp}(\Gamma_x^{\gamma(t)}(y + (1-t)a))\|^2 \\
&\leq \frac{\|a\|^2}{\|\tilde{z}\|^2} \|\mathcal{P}_{a^\perp}(\Gamma_x^{\gamma(t)}((1-t)a)) + \mathcal{P}_{a^\perp}(\Gamma_x^{\gamma(t)}y)\|^2 \\
&= \frac{\|a\|^2}{\|\tilde{z}\|^2} \|\mathcal{P}_{a^\perp}(\Gamma_x^{\gamma(t)}y)\|^2 \\
&\leq \frac{\|a\|^2 \|y\|^2}{\|\tilde{z}\|^2}.
\end{aligned}
$$

119   So

$$\|J_t^{\mathrm{norm}}(1)\| \leq \|J_t^{\mathrm{norm}}(0)\|c_1(K)\rho_t^2$$
$$\leq \frac{\|a\| \cdot \|y\|}{\|\tilde{z}\|}c_1(K)\|\tilde{z}\|^2$$
$$\leq c_1(K)\|a\| \cdot \|y\|(\|a\| + \|y\|),$$

120   and

$$d(\mathrm{Exp}_x(y + a), \mathrm{Exp}_z(\Gamma_x^z y)) \leq c_1(K)\|a\| \cdot \|y\|(\|a\| + \|y\|).$$

121   □

## B.4   Auxilliary lemmas

123   In the proofs of Lemma 1 and Lemma 2 we needed numerous auxiliary lemmas we are stating here.

124   We needed the following lemma which shows that both the exponential map and its inverse are
125   Lipschitz.

126   **Lemma 4.** *Let $x, y, z \in M$, and the distance of each two points is no bigger than $R$. Then under*
127   *Assumption 3*

$$(1 + c_2(K)R^2)^{-1}d(y, z) \leq \|\mathrm{Exp}_x^{-1}(y) - \mathrm{Exp}_x^{-1}(z)\| \leq (1 + c_3(K)R^2)d(y, z).$$

128   Intuitively this lemma relates the norm of the difference of two vectors of $\mathcal{T}_x\mathcal{M}$ to the distance
129   between the corresponding points on the manifold $\mathcal{M}$ and follows from bounds on the Hessian of the
130   square-distance function (Sakai, 1996, Ex. 4 p. 154).

131   *Proof.* The upper-bound is directly proven in Karcher (1977, Proof of Cor. 1.6), and we prove the
132   lower-bound via Lemma 3 in the supplement. Let $b = \mathrm{Exp}_y(\Gamma_x^y(\mathrm{Exp}_x^{-1}(z) - \mathrm{Exp}_x^{-1}(y)))$. Using
133   $d(y, b) = \|\mathrm{Exp}_y^{-1}(b)\|$ and Lemma 3,

$$d(y, z) \leq d(y, b) + d(b, \mathrm{Exp}_x(\mathrm{Exp}_x^{-1}(z)))$$
$$\leq \|\mathrm{Exp}_x^{-1}(y) - \mathrm{Exp}_x^{-1}(z)\|$$
$$+ c_1(K)\|\mathrm{Exp}_x^{-1}(y) - \mathrm{Exp}_x^{-1}(z)\|(\|\mathrm{Exp}_x^{-1}(y) - \mathrm{Exp}_x^{-1}(z)\| + \|\mathrm{Exp}_x^{-1}(y)\|)^2$$

134   □

135   The following contraction result is fairly classical and is proven using the Rauch comparison theorem
136   from differential geometry (Cheeger & Ebin, 2008).

137   **Lemma 5.** *(Mangoubi et al., 2018, Lemma 1) Under Assumption 3, for $x, y \in \mathcal{M}$ and $w \in T_x\mathcal{M}$,*

$$d(\mathrm{Exp}_x(w), \mathrm{Exp}_y(\Gamma_x^y w)) \leq c_4(K)d(x, y).$$

138   Eventually we need the following corollary of the famous Ambrose-Singer holonomy theorem (Am-
139   brose & Singer, 1953).

140   **Lemma 6.** *(Karcher, 1977, Section 6) Under Assumption 3, for $x, y, z \in \mathcal{M}$ and $w \in T_x\mathcal{M}$,*

$$\|\Gamma_y^z\Gamma_x^y w - \Gamma_x^z w\| \leq c_5(K)d(x, y)d(y, z)\|w\|.$$

## C   Proof of Lemma 7 and 8

142   In this section we prove two important lemmas from which the proof of our main result mainly comes
143   out. Then we show, in the last subsection, how to combine them to prove this main result.

144   **Lemma 7.** *Assume Assumptions 1, 2, 3 hold, and*

$$\epsilon \leq \min\left\{\frac{\hat{\rho}}{56\max\{c_2(K), c_3(K)\}\eta\beta}\log\left(\frac{d\beta}{\sqrt{\hat{\rho}\epsilon}\delta}\right), \left(\frac{\Im\hat{\rho}}{12\hat{c}\sqrt{\eta\beta}}\log\left(\frac{d\beta}{\sqrt{\hat{\rho}\epsilon}\delta}\right)\right)^2\right\} \quad (14)$$

145 *from the main theorem. There exists a constant $c_{\max}$, $\forall \hat{c} > 3, \delta \in (0, \frac{d\kappa}{e}]$, for any $u_0$ with $d(\tilde{x}, u_0) \le$*
146 *$2\mathscr{S}/(\kappa \log(\frac{d\kappa}{\delta}))$, $\kappa = \beta/\gamma$.*

$$T = \min\left\{ \inf_t \left\{ t | \tilde{f}_{u_0}(u_t) - f(u_0) \le -3\mathscr{F} \right\}, \hat{c}\mathscr{T} \right\},$$

147 *then $\forall \eta \le c_{\max}/\beta$, we have $\forall 0 < t < T$, $d(u_0, u_t) \le 3(\hat{c}\mathscr{S})$.*

148 **Lemma 8.** *Assume Assumptions 1, 2, 3 and Eq. (14) hold. Take two points $u_0$ and $w_0$ which*
149 *are perturbed from approximate saddle point, where $d(\tilde{x}, u_0) \le 2\mathscr{S}/(\kappa \log(\frac{d\kappa}{\delta}))$, $\text{Exp}_{\tilde{x}}^{-1}(w_0) -$*
150 *$\text{Exp}_{\tilde{x}}^{-1}(u_0) = \mu r e_1$, $e_1$ is the smallest eigenvector[1] of $H(\tilde{x})$, $\mu \in [\delta/(2\sqrt{d}), 1]$, and the algorithm*
151 *runs two sequences $\{u_t\}$ and $\{w_t\}$ starting from $u_0$ and $w_0$. Denote*

$$T = \min\left\{ \inf_t \left\{ t | \tilde{f}_{w_0}(w_t) - f(w_0) \le -3\mathscr{F} \right\}, \hat{c}\mathscr{T} \right\},$$

152 *then $\forall \eta \le c_{\max}/l$, if $\forall 0 < t < T$, $d(\tilde{x}, u_t) \le 3(\hat{c}\mathscr{S})$, we have $T < \hat{c}\mathscr{T}$.*

## C.1 Proof of Lemma 7

154 Suppose $f(u_{t+1}) - f(u_t) \le -\frac{\eta}{2}\|\text{grad} f(u_t)\|^2$.

$$
\begin{aligned}
d(u_{\hat{c}\mathscr{T}}, u_0)^2 &\le \left( \sum_{0}^{\hat{c}\mathscr{T}-1} d(u_{t+1}, u_t) \right)^2 \\
&\le \hat{c}\mathscr{T} \sum_{0}^{\hat{c}\mathscr{T}-1} d(u_{t+1}, u_t)^2 \\
&\le \eta^2 \hat{c}\mathscr{T} \sum_{0}^{\hat{c}\mathscr{T}-1} \|\text{grad} f(u_t)\|^2 \\
&\le 2\eta\hat{c}\mathscr{T} \sum_{0}^{\hat{c}\mathscr{T}-1} f(u_t) - f(u_{t+1}) \\
&= 2\eta\hat{c}\mathscr{T}(f(u_0) - f(u_{\hat{c}\mathscr{T}})) \\
&\le 6\eta\hat{c}\mathscr{T}\mathscr{F} = 6\hat{c}\mathscr{S}^2.
\end{aligned}
$$

## C.2 Proof of Lemma 8

156 Note that, for any points inside a region with diameter $R$, under the assumption of Lemma 8, we have
157 $\max\{c_2(K), c_3(K)\}R^2 \le 1/2$.

158 Define $v_t = \text{Exp}_{\tilde{x}}^{-1}(w_t) - \text{Exp}_{\tilde{x}}^{-1}(u_t)$, let $v_0 = e_1$ be the smallest eigenvector of $H(\tilde{x})$, then let $\hat{y}_{2,t}$
159 be a unit vector, we have

$$
\begin{aligned}
v_{t+1} &= (I - \eta H(\tilde{x}))v_t + C(K, \rho, \beta)d(u_t, w_t) \\
&\quad \cdot (d(u_t, \tilde{x}) + d(w_t, \tilde{x}) + d(\tilde{x}, u_0))\hat{y}_{2,t}.
\end{aligned}
\tag{16}
$$

160 Let $C := C(K, \rho, \beta)$. Suppose lemma 8 is false, then $0 \le t \le T$, $d(u_t, \tilde{x}) \le 3\hat{c}\mathscr{S}$, $d(w_t, \tilde{x}) \le 3\hat{c}\mathscr{S}$,
161 so $d(u_t, w_t) \le 6\hat{c}\mathscr{S}$, and the norm of the last term in Eq. (16) is smaller than $14\eta C\hat{c}\mathscr{S}\|v_t\|$.

162 Lemma 4 in the main paper indicates that

$$\|v_t\| \in [1/2, 2] \cdot d(u_t, w_t) = [3/2, 6] \cdot \hat{c}\mathscr{S}. \tag{17}$$

163 Let $\psi_t$ be the norm of $v_t$ projected onto $e_1$, the smallest eigenvector of $H(0)$, and $\varphi_t$ be the norm of
164 $v_t$ projected onto the remaining subspace. Then Eq. (16) is

$$
\begin{aligned}
\psi_{t+1} &\ge (1 + \eta\gamma)\psi_t - \mu\sqrt{\psi_t^2 + \phi_t^2}, \\
\phi_{t+1} &\le (1 + \eta\gamma)\phi_t + \mu\sqrt{\psi_t^2 + \phi_t^2}.
\end{aligned}
$$

165    Prove that for all $t \leq T$, $\phi_t \leq 4\mu t \psi_t$. Assume it is true for $t$, we have

$$4\mu(t+1)\psi_{t+1} \geq 4\mu(t+1) \cdot \left((1+\eta\gamma)\psi_t - \mu\sqrt{\psi_t^2 + \phi_t^2}\right),$$

$$\phi_{t+1} \leq 4\mu t(1+\eta\gamma)\phi_t + \mu\sqrt{\psi_t^2 + \phi_t^2}.$$

166    So we only need to show that

$$(1 + 4\mu(t+1))\sqrt{\psi_t^2 + \phi_t^2} \leq (1+\eta\gamma)\psi_t.$$

167    By choosing $\sqrt{c_{\max}} \leq \frac{1}{56\hat{c}^2}$ and $\eta \leq c_{\max}/\beta$, we have

$$4\mu(t+1) \leq 4\mu T \leq 4\eta C \mathscr{S} \cdot 14\hat{c}^2 \mathscr{T} = 56\hat{c}^2 \frac{C}{\hat{\rho}}\sqrt{\eta\beta} \leq 1.$$

168    This gives

$$4(1+\eta\gamma)\psi_t \geq 2\sqrt{2\psi_t^2} \geq (1 + 4\mu(t+1))\sqrt{\psi_t^2 + \phi_t^2}.$$

169    Now we know $\phi_t \leq 4\mu t \psi_t \leq \psi_t$, so $\psi_{t+1} \geq (1+\eta\gamma)\psi_t - \sqrt{2}\mu\psi_t$, and

$$\mu = 14\hat{c}\eta C \mathscr{S} \leq 14\hat{c}\sqrt{c_{\max}}\eta\gamma C \log^{-1}(\frac{d\kappa}{\delta})/\hat{\rho} \leq \eta\gamma/2,$$

170    so $\psi_{t+1} \geq (1+\eta\gamma/2)\psi_t$.

171    We also know that $\|v_t\| \leq 6\hat{c}\mathscr{S}$ for all $t \leq T$ from Eq. (17), so

$$6\hat{c}\mathscr{S} \geq \|v_t\| \geq \psi_t \geq (1+\eta\gamma/2)^t \psi_0$$

$$= (1+\eta\gamma/2)^t \frac{\mathscr{S}}{\kappa} \log^{-1}(\frac{d\kappa}{\delta})$$

$$\geq (1+\eta\gamma/2)^t \frac{\delta\mathscr{S}}{2\sqrt{d}\kappa} \log^{-1}(\frac{d\kappa}{\delta}).$$

172    This implies

$$T < \frac{\log(12\frac{\kappa\sqrt{d}}{\delta}\hat{c}\log(\frac{d\kappa}{\delta}))}{2\log(1+\eta\gamma/2)}$$

$$\leq \frac{\log(12\frac{\kappa\sqrt{d}}{\delta}\hat{c}\log(\frac{d\kappa}{\delta}))}{\eta\gamma}$$

$$\leq (2 + \log(12\hat{c}))\mathscr{T}.$$

173    By choosing $\hat{c}$ such that $2 + \log(12\hat{c}) < \hat{c}$, we have $T \leq \hat{c}\mathscr{T}$, which finishes the proof.

### 174    C.3    Proof of function value decrease at an approximate saddle point

175    With Lemma 7 and 8 proved, we can lower bound the function value in $O(\mathscr{T})$ iterations
176    decrease by $\Omega(\mathscr{F})$, thus match the convergence rate in the main theorem. Let $T' :=$
177    $\inf_t \left\{t | \tilde{f}_{u_0}(u_t) - f(u_0) \leq -3\mathscr{F}\right\}$. Let $\breve{}$ denote the operator $\mathrm{Exp}_{u_0}^{-1}(\cdot)$. If $T' \leq T$,

$$f(u_{T'}) - f(u_0)$$

$$\leq \nabla f(u_0)^T (u_{T'} - u_0) + \frac{1}{2}H(u_0)[\tilde{u}_{T'} - u_0, \breve{u}_{T'} - u_0]$$

$$+ \frac{1}{2}(\Gamma_{\tilde{x}}^{u_0} H(\tilde{x})\Gamma_{u_0}^{\tilde{x}} - H(u_0))[\tilde{u}_{T'} - u_0, \breve{u}_{T'} - u_0]$$

$$+ \frac{\rho}{6}\|\breve{u}_{T'} - u_0\|^3$$

$$\leq \tilde{f}_{u_0}(u_t) - f(u_0) + \rho d(u_0, \tilde{x})\|\breve{u}_{T'} - u_0\|^2$$

$$\leq -3\mathscr{F} + O(\rho\mathscr{S}^3) \leq -2.5\mathscr{F}.$$

178    If $T' > T$, then $\inf_t \left\{t | \tilde{f}_{w_0}(w_t) - f(w_0) \leq -3\mathscr{F}\right\} \leq T$, and we know $f(w_T) - f(w_0) \leq -2.5\mathscr{F}$.

**Remark 1.** *What is left is bounding the volume of the stuck region, to get the probability of getting out of the stuck region by the perturbation. The procedure is the same as in Jin et al. (2017). We sample from a unit ball in $\mathcal{T}_x\mathcal{M}$, where $x$ is the approximate saddle point. In Lemma 7 and 8, we study the inverse exponential map at the approximate saddle point $x$, and the coupling difference between $\mathrm{Exp}_x^{-1}(w)$ and $\mathrm{Exp}_x^{-1}(u)$. The iterates we study and the noise are all in the tangent space $\mathcal{T}_x\mathcal{M}$ which is a Euclidean space, so the probability bound is same as the one in Jin et al. (2017).*

## D   Experiment with retraction

In the main algorithm and its proof, we use the exponential map in the algorithm. The exponential map is easy to compute for many manifolds, but one may also use *retraction* as a first order approximation of exponential map. We do not theoretically study retraction, but the experiment below shows that replacing exponential by a smooth retraction works well practically.

(a)                                                       (b)

Figure 4: (a) Function $f$ with saddle point on a sphere. $f(x) = x_1^2 - x_2^2 + 4x_3^2$. We plot the contour of this function on unit sphere. The main algorithm initializes at $x_0 = [1, 0, 0]$ (a saddle point), perturbs it towards $x_1$ and runs Riemannian gradient descent, and terminates at $x^* = [0, -1, 0]$ (a local minimum). We amplify the first iteration to make saddle perturbation visible. (b) We replace exponential map by retraction $R_x(v) = (x + v)/\|x + v\|_2$ and do the same experiment, which addresses the generality of the result. We do not provide in this paper proof for algorithm with retraction, but practically the iterates converge to an approximate saddle point.

## Footnotes

[1]"smallest eigenvector" means the eigenvector corresponding to the smallest eigenvalue.