[Reviews · NeurIPS 2019]

Reviewer 1



Essentially the paper states an algorithm page 3 and the rest of the paper is based on explaining the "proof strategy", i.e., the authors explain the various results which are used to build the main convergence-rate theorem. I wonder how many NeurIPS readers will be able to get anything out of pages 3-7 which describes results from geometry. While the result might be of theoretical significance, the algorithm requires knowledge of the exponential map which is usually a very complicated operator. The main non-trivial manifolds for which the exponential map is actually known are the Grassmannian and Stiefel manifolds, hence I wonder if the authors should have just focused on these two manifolds which would considerably simplify the paper; and by specialising to these the results can probably be improved. I personally don't see much point in describing more general results (unless it's aimed at a more mathematical audience or in the appendices) unless the authors can generalise their results to more general retractions, which would improve the applicability of the results.

Reviewer 2



In this paper the authors propose an algorithm for optimization over manifolds, capable of converging to second-order stationary points, with a rate of convergence of 1/eps^2 The presentation of the paper in general, and the results in particular, is quite clear. I particularly really liked the introduction (despite some typos), and the review of existing work and results. The proofs seem to be correct, although I didn't follow carefully the supplementary material. I appreciate the presence of examples, but the analysis there is somehow poor. What about the complexity of each manifold? How does this affect the constants in your results? What about the rate of convergence? In the Reproducibility Checklist answers, the authors say that the code is available for dowloading. However, I couldn't find it in the paper. Minor comments: There are some typos and weird sentences, specially in the introduction. For instance - "Riemnnian manifolds, but not clear about the complexity..." - escapes (an approximate) saddle pointS Also in the footnote 2, it doesn't specify that \lambda_min is the eigenvalue of the Hessian. I think that the enumeration in page 5 (after Lemma 1) is not necessary.

Reviewer 3



The authors analyse gradient methods on Riemannian manifolds and prove that their suggested algorithm escapes saddle points with a specific rate. This work was technical but clearly written. The quality is good. The work is significant. Only limited experiments were conducted.

[Author Response · NeurIPS 2019]

Thank you for all your comments and corrections; they will be incorporated into our revised version.

**All reviewers.** The paper focuses on theoretical guarantees, however we agree providing more experiments can be helpful. We plan to add (a) An example using general retraction—to show the generality of our approach (see an illustration below); (b) to compare the algorithm's rate on two manifolds with different curvature—to illustrate how curvature affects the rate empirically.

Our existing experiments illustrate that the iterates escape from saddle points and converge fast to an approximate second order stationary point. Figure 1 in the paper shows that, after perturbation, the iterates escape the saddle point $x_0$ and converges to $x^*$.

**Reviewer 1.**

1. The ML community has shown growing interest in manifold optimization and rates of escape from saddle points; papers on both non-convex optimization (e.g., Jin et al., 2017) and Riemannian optimization (e.g., Bécigneul & Ganea, 2019) are published in top ML conferences.

2. If we agree about the importance of Grassmannian and Stiefel manifolds, numerous other interesting manifolds also have closed-form exponential maps: the *Minkowski space, the hyperbolic space, $SE(n)$, $SO(n)$*...(see, Miolane et al. (2018); Boumal et al. (2014) and their open-source packages and Bécigneul & Ganea (2019, Sec 5) for an example in NLP). Focusing on the Grassmanian and Stiefel manifolds would be too restrictive and would leave out all these other example applications.

3. We stress that a major contribution is to derive convergence rate for the Riemannian gradient expressed in terms of *the manifold curvature*. As one may already see from the "proof strategy", this is made possible by involved results in differential geometry which control the evolution of the exponential map in a curvature dependent manner (Rauch comparison theorem on Jacobi fields).

As far as we are aware, there is no such results for general retraction, and convergence results for general retractions are therefore not curvature dependent and consequently *not* truly geometric. Thereby convergence rate in Riemannian optimization when retraction is used, (see, e.g., Criscitiello & Boumal, 2019; Lee et al., 2017; Zhang & Zhang, 2018) is based on the Lipschitz constants of the pullback function instead, which are hard to quantify and mix the properties of the function and the manifold.

However, as explained in Section 3, our algorithm also works with a general retraction. We agree about providing additional experiments to illustrate this feature. We re-did Figure 1 in the paper by replacing the *exponential map* with the *retraction $R_x(v) = (x + v)/\|x + v\|_2$*; we show the resulting figure here. We observe that the behavior is very similar as when the exponential map is used.

4. In the theoretical part of the paper, we state the key technical challenges, main lemmas and the roadmap/proof sketch to the final result, in order to give the reader the intuition behind the proofs. This certainly interests a part of the community, and also strikes a balance between giving full details and completely skipping all proof in the main body.

**Reviewer 2.** We thank the reviewer for their helpful comments. We will correct the small typos (thanks!). We will expand the discussion on our examples and empirical results to (a) explore the dependence of the rate on the geometric and function-dependent constants for the specific examples studied, and (b) empirically investigate whether for a manifold more curved than a sphere (large $|K|$), the convergence rate is slower than on the sphere. We will also post our code that implements the algorithm in Matlab for the examples in our paper later.

# References

Bécigneul, G. and Ganea, O.-E. Riemannian adaptive optimization methods. In *ICLR*, 2019.

Boumal, N., Mishra, B., Absil, P.-A., and Sepulchre, R. Manopt, a Matlab toolbox for optimization on manifolds. *JMLR*, 2014.

Criscitiello, C. and Boumal, N. Efficiently escaping saddle points on manifolds. *ArXiv:1906.04321*, 2019.

Jin, C., Ge, R., Netrapalli, P., Kakade, S., and Jordan, M. I. How to escape saddle points efficiently. In *ICML*, 2017.

Lee, J. D., Panageas, I., Piliouras, G., Simchowitz, M., Jordan, M. I., and Recht, B. First-order methods almost always avoid saddle points. *ArXiv:1710.07406*, 2017.

Miolane, N., Mathe, J., Donnat, C., Jorda, M., and Pennec, X. Geomstats: a python package for Riemannian geometry in machine learning. *ArXiv:1805.08308*, 2018.

Zhang, J. and Zhang, S. A cubic regularized Newton's method over Riemannian manifolds. *ArXiv:1805.05565*, 2018.


[Meta-Review · NeurIPS 2019]

The paper considers the problem of escaping saddle points for optimization problems on certain manifolds. There are theoretical contributions that are valuable.